# Spiritual boredom is associated with over- and underchallenge, lack of value, and reduced motivation
Thomas Goetz [1] ✉, Jonathan Fries [1], Lisa Stempfer [1], Lukas Kraiger [1], Sarah Stoll [1], Lena Baumgartner[1], Yannis L. Diamant[1], Caroline Porics[1], Bibiana Sonntag[1], Silke Würglauer[1], Wijnand A. P. van Tilburg [2] & Reinhard Pekrun [2,3,4]

The emotion of boredom has attracted considerable research interest. However, boredom experienced in spiritual contexts (i.e., spiritual boredom) has rarely been investigated. Based on control-value theory (CVT), we investigated the occurrence, antecedents, and motivational effects of spiritual boredom in five different spiritual contexts: yoga, meditation, silence retreats, Catholic sermons, and pilgrimage. For each context, we conducted two independent studies, one including trait and another including state measures. The set of 10 studies included a total sample of $N = 1267$ adults. We complemented individual study results with an internal meta-analysis. The results showed a mean level of spiritual boredom of $\bar{M} = 1.91$ on a scale of 1 to 5. In line with CVT, spiritual boredom was positively related to being overchallenged ($\bar{r} = 0.44$) in 9 out of the 10 studies and positively related to being underchallenged ($\bar{r} = 0.44$) in all studies. Furthermore, as expected, spiritual boredom was negatively related to perceived value in all studies ($\bar{r} = -0.54$). Finally, boredom was negatively related to motivation to engage in spiritual practice ($\bar{r} = -0.46$) across studies. Directions for future research and practical implications are discussed.

*"The existential vacuum manifests itself mainly in a state of boredom"*.

(Viktor E. Frankl; Austrian neurologist, psychologist, and philosopher; 1905–1997[1])

The last 15 years have seen a substantial increase in studies on boredom[2]. A key reason for this growing interest is the accumulating empirical evidence on the effects of boredom on a wide range of critically important outcomes, including poorer physical and mental health, problematic eating habits, substance use, reduced motivation, and poor performance[3].

One context that is becoming increasingly important in Western societies and can cause boredom is the spiritual context[4], which includes practices such as yoga, meditation, and pilgrimage[5,6]. It is intriguing that there is a clear lack of research on the levels, antecedents, and effects of boredom in spiritual contexts. There is a study[7] that investigated "void" (a combination of boredom and psychological entropy), from which one could draw a cautious and limited conclusion about the levels, antecedents and effects of spiritual boredom. An example of the neglect of spiritual boredom

is the highly cited "Handbook of Psychology of Religion and Spirituality"[8], which does not even mention the terms "boredom" or "bored".

A key reason why spiritual boredom has been neglected may be that it is theoretically unexpected that spiritual contexts could ever be boring. People typically engage in these contexts voluntarily and with high motivation, seeking meaning and fulfillment in their lives. The intuition that spiritual boredom is likely to be low is consistent with propositions of Pekrun's[9–12] control-value theory (CVT). Spiritual practices are intuitively neither over- nor underchallenging, which means that levels of perceived control are likely to be appropriate. Furthermore, spiritual contexts can be assumed to be inherently high in value. According to CVT, appropriate levels of control and high perceived value should prevent boredom[13,14].

However, a closer look reveals that some practices are not individualized (e.g., sermons, instructed yoga practices[7]). As such, they could lead to being over- or underchallenged and, consequently, to boredom. Furthermore, some spiritual practices may actually be of little value to some people, especially if they become routine (e.g., meditation; walking routines on long

[1]Department of Developmental and Educational Psychology, Faculty of Psychology, University of Vienna, Vienna, Austria. [2]Department of Psychology, University of Essex, Essex, UK. [3]Institute for Positive Psychology and Education, Australian Catholic University, Sydney, Australia. [4]Department of Psychology, Ludwig-Maximilians-Universität München, München, Germany. ✉e-mail: thomas.goetz@univie.ac.at

pilgrimages) or are performed due to externally defined obligations (e.g., attending religious services), which also might lead to boredom.

In this research, we drew upon the theoretical propositions on boredom outlined in CVT and explored the extent to which boredom is experienced during spiritual practice, and whether it is related in theoretically plausible ways to the antecedents of boredom mentioned above (non-optimal levels of control, i.e., over- or underchallenge, and value appraisals). In addition, we examined how boredom relates to the motivation to engage in spiritual practice, also within the framework of CVT. We examined these relations using both trait and state measures to capture both habitual and situational experiences of spiritual boredom. To gain an understanding of boredom in spiritual practice, we selected five different exemplary contexts: Yoga, meditation, silence retreats, sermons (as part of Catholic worship), and pilgrimages. Based on the results from the 10 single studies included in this research, we conducted internal meta-analyses on the levels, antecedents, and effects of spiritual boredom.

As a first step, we reviewed the current state of research on definition, occurrence, antecedents, and effects of spiritual boredom. To this end, we conducted a comprehensive literature search in PsycInfo, APA PsycArticles, PSYNDEX, and Web of Science. We searched for publications on "spiritual boredom" as well as publications on boredom in relation to the five specific spiritual contexts that are the focus of our research: yoga, meditation, sermons, silence retreats, and pilgrimage. We used the search terms "(Boredom OR Bored) AND (Spiritual* OR Yoga OR Meditation OR Meditating OR Sermon OR Church OR 'Silence Retreat' OR Pilgrimage)."

The search was performed on March 26th, 2024, and yielded a total of 318 results after removing duplicates. In the next step, we screened the publications based on the following criteria: 1) providing a definition of spiritual boredom, and/or 2) reporting the prevalence of boredom experienced in the spiritual context, and/or outlining or empirically investigating 3) antecedents or 4) effects of spiritual boredom. None of the articles retrieved met any of these criteria. For example, the studies we found looked at whether spiritual or religious people were less bored than others[15].

Therefore, it is clear that there is currently a lack of empirical research on spiritual boredom. However, beyond this literature review, we identified a few relevant studies that are related to spiritual boredom, although not directly focusing on it. These include a study investigating "void"[7] (a combination of boredom and psychological entropy) and a publication in the field of theology[16] assessing boredom during Catholic services as an additional variable that was not central to his study.

Depending on historical time and culture, different terms have been used to describe the feeling of "boredom," including tedium, melancholia, acedia, ennui, and monotony[17]. Beyond these different terms, there are several definitions of boredom that can vary across scientific domains. To conceptualize boredom in our study, we refer to the component process model of emotions[18,19], which posits that emotions are best understood through their underlying processes. From this perspective, boredom can be defined as a unique emotional process consisting of four components: affective (an unpleasant, aversive feeling), cognitive (altered perception of time, mind wandering, attention failures), motivational (a desire to withdraw from the current situation), and physiological/expressive (low arousal, yawning, looking tired[3,20,21]).

Similar to other types of boredom (e.g., academic boredom, leisure time boredom), spiritual boredom can be conceptualized as either a trait or a state. This distinction aligns with previous research on boredom and other emotions (e.g., anxiety[22,23]). Trait spiritual boredom is defined as habitual boredom experienced in spiritual situations, that is, boredom that recurs across various spiritual contexts and over time. In contrast, state spiritual boredom refers to the current experience of boredom in a specific spiritual situation. Based on the relative universality assumptions of CVT[9–11,24], similar structural relations with antecedents and outcomes can be assumed for both trait and state spiritual boredom.

An important issue in defining spiritual boredom is determining what constitutes a "spiritual context." Definitions of the term "spirituality" vary widely across different fields of research, including psychology, sociology,

philosophy, theology, cultural studies, and history[25]. Despite these variations, a common element in almost all definitions of spirituality is the search for and belief in something sacred that transcends the material world[26–28]. Consistent with this view, the Cambridge Dictionary defines "spirituality" as "the quality that involves deep feelings and beliefs of a religious nature, rather than the physical parts of life."

However, whether a situation is considered spiritual can vary substantially between and within individuals. Everyday experiences can sometimes be perceived as spiritual (e.g., looking at the night sky). Conversely, what might seem like an obvious spiritual context, such as the routine recitation of prayers, may be perceived as non-spiritual (e.g., as the mere fulfillment of duty). In our work, we follow approaches used in the study of other types of boredom (e.g., "test boredom"[29] and "academic boredom" in academic settings[3]) and define spiritual boredom as boredom experienced in situations that are typically considered to be spiritual in nature.

Spiritual boredom differs from other types of boredom (e.g., academic boredom) in terms of (a) the population experiencing it, which includes people who seek spiritual development and often search for greater meaning in life; (b) the settings, which typically are "silent" environments where spirituality can be experienced and where visits usually are voluntary; and (c) its consequences in terms of a reduction in motivation for spiritual practice and, consequently, spiritual growth.

Based on our literature search, we were unable to identify any studies that assessed the levels of boredom experienced in spiritual contexts. Beyond our literature search, we identified one study[7] investigating "void" in the context of mindfulness meditation. Void is a construct that is related to boredom, but it is limited to situations in which nothing is perceived to be happening. The levels of void in mindfulness meditation ranged between 2.4 and 2.8 on a scale ranging from 1 (*completely disagree*) to 7 (*completely agree*).

Another quantitative, albeit weakly operationalised, indicator of high levels of spiritual boredom was reported in another study[16]. In a sample of German Catholics ($N = 2649$), approximately 50% believed that boredom in Catholic services was particularly prevalent during the sermon. Additionally, several qualitative studies provide statements that can be interpreted as depicting experiences of boredom in spiritual contexts. For example, Cassaniti notes a sentiment regarding a sermon in a Buddhist context: "When I go to an Asanha Bucha Day sermon, I feel … bored"[30].

Beyond empirical research, boredom in spiritual contexts is often described in the press and social media. For example, there are articles discussing boring sermons, often featuring statements from worshippers, such as: *"I quite like the whole liturgy, but this impression is ruined again by the boring sermon. In my opinion, the entire sermon consists of empty words"*[31]. There are numerous indicators of boredom in Christian traditions, such as paintings depicting people sleeping during sermons[25]. In the Middle Ages, boredom was recognized as a spiritual malaise known as "acedia" (Latin word), characterized by listlessness and melancholy[32]. Christians referred to it as the "demon of noontide," a concept described by St. Thomas Aquinas (1273) as the "sorrow of the world" and the "enemy of spiritual joy"[15,33,34]. In summary, although empirical evidence on the extent of spiritual boredom is lacking, there is ample anecdotal evidence from the Middle Ages to the present that spiritual boredom may indeed be common.

Although there is no specific theory addressing the antecedents of spiritual boredom, it is reasonable to assume that the primary theoretical antecedents of boredom might also apply to spiritual boredom. These primary antecedents include attentional processes[35], cognitive appraisals[20], perceived meaning[36], and functional value[37].

Two constructs frequently highlighted in theories of boredom are inappropriate levels of control (both very high and very low) and lack of value. These constructs are of central importance in the control-value theory (CVT[9,12,38]). Originally developed primarily for the context of achievement emotions, CVT is increasingly being used to explain other emotions[12] and contexts (e.g., leadership[39]).

**Fig. 1 | Antecedents and effects of spiritual bore-dom.** *Note.* Hypotheses H1, H2, H3, and the exploratory research question are presented with +/− indicating positive vs. negative relations.

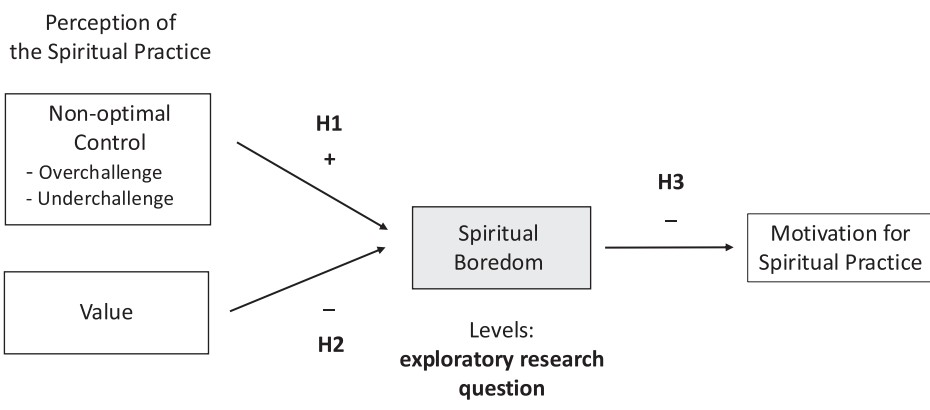

CVT posits that individuals' perceptions of their personal control over, and the value of activities and outcomes are the most important proximal psychosocial antecedents of boredom. CVT includes assumptions on the universality of boredom antecedents[24]. According to the theory, the structural relations between boredom and its antecedents are similar across different contexts, including spiritual ones. Nevertheless, as the characteristics of spiritual situations can be quite different from other situations (e.g., academic learning or testing situations), for example in terms of the level of competitiveness, these situational differences might lead to different levels of spiritual boredom compared to boredom experienced in other contexts.

Perceived control refers to individuals' perceived causal influence over their actions and outcomes[40]. According to CVT, the relation between spiritual boredom and perceived control is curvilinear, with higher levels of boredom experienced when perceived control is either very low or very high[3,38]. This aligns with traditional approaches to boredom, which attribute its occurrence to a misfit between the person and the environment[14,41,42].

The proposed link between levels of control and boredom has received partial support in studies on academic boredom. Perceived control has mainly been found to negatively relate to boredom[20,21,38,43,44]. This may be because tasks in educational settings are rarely designed to be extremely easy, resulting in very high levels of perceived control and underchallenge being uncommon[45–47]. Furthermore, traditional self-report scales assessing perceived control may not be appropriate for assessing very high and low levels of control. Thus, recent studies have used direct reports of being overchallenged and underchallenged as indicators of very low and very high levels of perceived control. These studies have shown that boredom can indeed occur in situations characterized by both very high and very low perceived control[29,48].

Perceived value refers to the perceived relevance and meaning of actions and outcomes to an individual[9,12]. According to CVT, there is a negative relation between perceived value and boredom. Given the universality assumptions of CVT, we expect that this relation will also apply to spiritual boredom. In this respect, spiritual boredom (like boredom in other contexts) contrasts with other emotions that are generally amplified by perceived importance[24]. Similar to boredom in other contexts, spiritual boredom is expected to be negatively related to all facets of value (e.g., intrinsic value, extrinsic value[12,49]). Empirical studies outside the spiritual context have consistently reported negative correlations between boredom and various types of perceived value[20,46,50]. In particular, value provided by 'meaning' is often highlighted in the literature as a crucial antecedent of boredom[1,15].

In the context of spiritual practice, we identified a study that examined the relationship between perceived value and experiences related to spiritual boredom. This study of novice meditators[7] ($N = 175$) found that the "void" experienced during mindfulness meditation was negatively related to the perceived meaning of meditation ($r = -0.49$; $p < 0.001$). This finding is clearly consistent with the assumptions of CVT.

CVT also provides a framework for understanding the effects of spiritual boredom. Spiritual boredom may deplete cognitive resources by causing mind wandering. For instance, individuals may struggle to maintain focus on bodily sensations during yoga practice. This depletion may subsequently reduce motivation to engage in spiritual tasks, leading to behaviors such as postponing meditation sessions or only very briefly engaging in meditation. Boredom may also encourage the use of superficial strategies, such as not reflecting deeply during a sermon. Furthermore, boredom may interfere with the flexible adaptation of strategies, such as failing to correct inefficient body movements during yoga, ultimately leading to diminished performance and reduced spiritual growth. In essence, spiritual boredom can lead to a number of negative outcomes that hinder the effectiveness of spiritual practices and impede personal development.

Existing research beyond the spiritual context supports theoretical assumptions on the effects of boredom[20,21,50–54]. In our study, we focus on one critical effect of spiritual boredom: its impact on motivation to engage in spiritual practices.

We found a study indicating that spiritual boredom is related to outcome variables. This research[7] found that "void" was negatively related to engagement in meditation ($r = -0.54$; $p < 0.001$). This finding is consistent with CVT assumptions that experiences of boredom lead to a loss of motivation. Beyond this study, there are scattered findings on the motivational effects of negative emotions other than boredom (e.g., anxiety, stress, worry) experienced during spiritual practices. These studies indicate that such negative emotions can act as barriers to spiritual practice and as predictors of attrition[55–58]. Boredom may have a similarly negative impact on motivation for spiritual practices. Thus, spiritual boredom may represent a previously unrecognized indicator or form of spiritual struggle, with research on the topic emphasizing experiences of tension, strain, or conflict in relation to religion and spirituality[59,60].

In sum, based on CVT[12], it can be assumed that perceptions of being over- or underchallenged during spiritual practices, as well as judgments that these practices are not valuable, should lead to higher levels of spiritual boredom. Given that over- and underchallenge as well as low perceived value is likely to occur during spiritual practices (e.g., underchallenging sermons; sermons that have no relevance to daily life), it follows that boredom may be prevalent in such contexts. Furthermore, spiritual boredom is expected to lead to low motivation for subsequent practice. In our literature search, we did not find any studies that specifically addressed the occurrence, antecedents, and effects of spiritual boredom. However, as outlined above, various scattered findings from studies indirectly related to our research questions support these assumptions.

Based on the theoretical propositions of CVT, we aimed to test the following hypotheses (see also Fig. 1). We expect these hypotheses to apply to both trait and state spiritual boredom.

*Exploratory Research Question*: What are the levels of boredom in different spiritual contexts?

*Hypothesis 1*: Spiritual boredom shows significant positive relations with perceptions of being over- or underchallenged in spiritual practice.

*Hypothesis 2*: Spiritual boredom shows significantly negative relations with the perceived value of spiritual practice.

*Hypothesis 3:* Spiritual boredom shows significantly negative relations with the motivation to engage in spiritual practice.

## Methods

We investigated five different spiritual contexts: yoga, meditation, silence retreats, sermons (as part of Catholic services), and pilgrimage. By studying five different contexts, we aimed to test the generalizability of our findings. For each context, we conducted two studies with fully independent samples, one using trait measures and the other using state measures of spiritual boredom. The 10 studies were paralleled as much as possible with respect to the assessed constructs to ensure comparability. Based on the 10 studies, we conducted an internal meta-analysis to provide a concise synthesis of our findings. None of the studies were preregistered.

### Spiritual contexts

Our study focused on five traditional spiritual practices with different religious/spiritual roots, but commonly observed in Western societies. We wanted to assess contexts that were different, but not too different, as a starting point for research into spiritual boredom.

One focus was on (1) *yoga*, a spiritual practice that has existed for at least 2500 years[61] and pursues a unifying experience of body and mind. We also focused on (2) *meditation*, which also has existed for at least 2500 years. The key spiritual aspect of meditation is to train attention and awareness to achieve a mentally clear and emotionally calm and stable state. As for (3) *silence retreats*, the practice of consciously remaining silent for spiritual reasons probably dates back to the earliest humans and has a long tradition in many religions, such as Buddhism, Hinduism, and Christianity. Listening to (4) *sermons* as part of Catholic services reflects a spiritual practice that has existed for about 2000 years. We have explicitly not referred to Catholic services as a whole, as they consist of different elements (e.g., prayers, chants, rituals, communion, silence) that may evoke different levels of boredom. Finally, (5) *pilgrimage* is also a traditional spiritual practice found in many religions, such as Christianity, Hinduism, and Islam. However, it is also practiced outside religious contexts, often as a form of spiritual walking or traveling in search of moral or spiritual meaning.

### Trait and state assessments

For each of our five spiritual contexts, we conducted both a trait and a state study. The trait assessments focused on habitual experiences of spiritual boredom, while the state assessments focused on situational, momentary experiences. In both types of assessment, we examined the antecedents of spiritual boredom and its impact on motivation to engage in spiritual practice.

### Participants

Table 1 provides an overview of the samples for all 10 studies on spiritual boredom in terms of sample sizes, gender distributions, and age. The language of the recruitment process as well as the questionnaires was German, resulting in samples of German-speaking participants. All 10 studies included in this work received ethical approval from the Institutional Review Board (IRB) of the Department of Developmental and Educational Psychology at the University of Vienna.

### Procedure

All 10 studies took place between December 2021 and July 2024. For all studies, participants were recruited through a variety of methods, including using existing contacts with practitioners of the spiritual practice and institutions offering such practices (e.g., yoga studios, pilgrim associations), and social networks such as WhatsApp, LinkedIn, and Facebook. Contacted individuals and institutions were asked to share information about the study (i.e., snowball sampling[62]), through which a link to an online questionnaire was distributed. In this recruitment process, studies were labeled as research investigating emotions in the respective context (i.e., yoga, meditation, silence retreat, Catholic sermons, pilgrimage).

For the trait assessments, we aimed to recruit participants with experience in the spiritual practice being studied (i.e., former and current

practitioners). For the state assessments, we sought participants who were currently practicing the respective spiritual practice. The questionnaires were created using the SoSci Survey platform[63]. Upon activation of the link, participants received detailed information about the study, data handling procedures, guarantees of full anonymity, and contact information for the researchers. An exception was the study on state sermon boredom (Study 8). In this study, participants were recruited by trained test administrators in four Catholic churches directly after attending a service.

In all 10 studies, participants had to be at least 18 years old to take part. Participants had to provide their consent before proceeding with the questionnaire. There were no exclusion criteria other than not having experience in spiritual practice, being under 18 years of age, and not giving consent. As for the state assessments, participants were asked to complete the questionnaires immediately after engaging in the spiritual practices. The questionnaire began with demographic information, followed by the assessment of all other variables. No data on race/ethnicity were collected. Participants could stop the assessment at any time without having to give a reason. There was no compensation for participating in the study. The average time taken to complete the questionnaires across the 10 studies ranged from 2.97 min (silence retreat—trait; Study 6) to 8.08 min (pilgrimage—state; Study 10).

### Measures

**Spiritual boredom.** We developed a total of 10 scales to assess trait and state boredom in each of the five spiritual contexts that were addressed in our research (see overview in Table 2; reliabilities of all spiritual boredom scales are shown in this table). Within each spiritual context, the content of the trait and state boredom items was completely parallel, and both scales contained the same number of items. As the spiritual contexts of yoga, meditation, silence retreat, and pilgrimage consist of different typical facets, we developed scales including items assessing boredom related to these facets based on an approach outlined in previous work[64], which suggests an assessment of situational facets (e.g., for yoga: physical experiences, breathing exercises, relaxation phase,…). However, as sermons do not contain such typical elements, we developed a scale for this context based on the Achievement Emotions Questionnaire (AEQ[50]), which takes into account different components of boredom as outlined earlier (i.e., affective, cognitive, motivational, and physiological/expressive components). In addition, each of the scales included an item that directly assessed the overall level of boredom. Examples for these items are: "*When I practice yoga, I usually get bored*" (Study 1), "*During the yoga session I had just completed, I was bored*" (Study 2).

Participants responded to the items in all boredom scales using a 5-point Likert scale, ranging from 1 (*completely disagree*) to 5 (*completely agree*). All 10 scales are documented in the online supplemental material (SM1).

To validate the scales, we included a well-established state boredom scale in the state assessments (Studies 2 and 4) for two spiritual contexts, namely yoga and meditation: the MSBS-SF[65], which is the short form of the Multidimensional State Boredom Scale (MSBS[66]). We adapted the items to suit our assessment, which took place immediately after the yoga and meditation sessions. Example items from the MSBS-SF include "*My mind was wandering*" and "*I was easily distracted*". Cronbach's alpha for this scale was $\alpha = 0.94$ for both contexts (for the full scale see SM1: SM1_11).

We found strong correlations between the MSBS-SF and the Yoga Boredom Scale—State (YBS-S; $r = 0.90$, $p < 0.001$) as well as the MSBS-SF and the Meditation Boredom Scale—State (MBS-S; $r = 0.70$, $p < 0.001$), indicating high convergent validity for our state spiritual boredom scales.

**Being over- and underchallenged.** Based on previous work[48], we developed scales to assess levels of over- and underchallenge in each of the 10 studies. Each of the scales was related to the facets of spiritual practices assessed by our spiritual boredom scales. Within each of the spiritual contexts, the scales for being overchallenged and underchallenged were parallel in content and consisted of the same number of items. Furthermore, within each spiritual context, the content of the trait and state

**Table 1 | Study Participants**

| Spiritual Context | Study | Trait/State | $N$ | Gender f/m/d | Age ($M$, $SD$) | Age (min; max) |
|---|---|---|---|---|---|---|
| Yoga | 1 | Trait | 159 | 138/20/1 | 37.41 (13.37) | 20; 71 |
| | 2 | State | 57 | 48/9/0 | 35.16 (11.43) | 22; 65 |
| Meditation | 3 | Trait | 63 | 50/13/0 | 28.71 (10.90) | 18; 67 |
| | 4 | State | 61 | 43/16/2 | 46.56 (15.61) | 19; 82 |
| Silence retreat | 5 | Trait | 90 | 53/34/3 | 46.20 (17.60) | 19; 83 |
| | 6 | State | 40 | 21/17/0 | 46.18 (15.33) | 26; 70 |
| Sermon | 7 | Trait | 414 | 127/226/9 | 37.55 (15.32) | 18; 78 |
| | 8 | State | 97 | 53/43/1 | 51.52 (17.92) | 18; 87 |
| Pilgrimage | 9 | Trait | 162 | 115/45/2 | 52.46 (13.88) | 19; 76 |
| | 10 | State | 124 | 95/29/0 | 47.89 (11.99) | 23; 71 |

In some cases, the counts relating to gender do not add up to the full sample size, due to missing data.
$N$ sample size, f/m/d female participants/male participants/diverse participants.

**Table 2 | Spiritual Boredom Scales**

| Spiritual Context | Study | Trait/State | Scale | $N_i$ | Sample item | α |
|---|---|---|---|---|---|---|
| Yoga | 1 | Trait | YBS-T | 7 | The physical exercises in yoga usually bore me. | 0.71 |
| | 2 | State | YBS-S | 7 | During the yoga session I had just completed, I was bored with the physical exercises. | 0.93 |
| Meditation | 3 | Trait | MBS-T | 6 | The concentration exercises in meditation usually bore me | 0.86 |
| | 4 | State | MBS-S | 6 | During the meditation session I had just completed I was bored with the concentration exercises. | 0.82 |
| Silence retreat | 5 | Trait | SRBS-T | 11 | I usually get bored in silence at silence retreats | 0.87 |
| | 6 | State | SRBS-S | 11 | I was bored with the silence. | 0.93 |
| Sermon | 7 | Trait | SBS-T | 7 | During the sermon, I have the feeling that time passes more slowly than usual | 0.93 |
| | 8 | State | SBS-S | 7 | During the sermon, I had the feeling that time passed more slowly than usual. | 0.96 |
| Pilgrimage | 9 | Trait | PBS-T | 11 | Walking on a pilgrimage usually bores me | 0.82 |
| | 10 | State | PBS-S | 11 | I was bored of walking. | 0.84 |

$N_i$ number of items, YBS-T Yoga Boredom Scale—Trait, YBS-S Yoga Boredom Scale—State, MBS-T Meditation Boredom Scale—Trait, MBS-S Meditation Boredom Scale—State, SRBS-T Silence Retreat Boredom Scale—Trait, SRBS-S Silence Retreat Boredom Scale—State, SBS-T Sermon Boredom Scale—Trait, SBS-S Sermon Boredom Scale—State, PBS-T Pilgrimage Boredom Scale—Trait, PBS-S Pilgrimage Boredom Scale—State.

challenge items was fully parallel, and the trait and state challenge scales included the same number of items.

Sample items for being over- and underchallenged for the yoga context are "*The breathing exercises usually overchallenge me*" and "*The breathing exercises usually underchallenge me*" (trait scales), and "*The breathing exercises overchallenged me*" and "*The breathing exercises underchallenged me*" (state scales). For all scales, participants responded using a 5-point Likert scale, ranging from 1 (*completely disagree*) to 5 (*completely agree*).

The number of items ($n_i$) and Cronbach's alpha for being over-/underchallenged ($\alpha_{o/u}$) in the trait and state assessments were as follows: for yoga $n_i = 5$, $\alpha_{o/u\_trait} = 0.81/0.88$ (Study 1) and $\alpha_{o/u\_state} = 0.81/0.91$ (Study 2); for meditation $n_i = 4$, $\alpha_{o/u\_trait} = 0.83/0.86$ (Study 3) and $\alpha_{o/u\_state} = 0.80/0.85$ (Study 4); for silence retreats $n_i = 10$, $\alpha_{o/u\_trait} = 0.75/0.92$ (Study 5) and $\alpha_{o/u\_state} = .83/.91$ (Study 6); for sermon $n_i = 1$ (Studies 7 and 8); for pilgrimage $n_i = 12$, $\alpha_{o/u\_trait} = .82/.87$ (Study 9) and $\alpha_{o/u\_state} = 0.79/0.81$ (Study 10). The items of all 20 scales are presented in the online supplemental material (SM2).

**Perceived value.** We developed scales assessing value based on previous work[49]. The scales were each related to the facets of the spiritual practices as assessed with the spiritual boredom scales. The items were fully parallel in content for the trait and state assessments, and thus the number of items within spiritual contexts was identical. For example, a sample item for the pilgrimage context is "*Walking on a pilgrimage is important to me*" (trait assessment) and "*Walking on the pilgrimage is important to me*" (state assessment). For all scales, participants responded using a 5-point Likert scale, ranging from 1 (*completely disagree*) to 5 (*completely agree*).

The number of items ($n_i$) and Cronbach's alpha for the value scales in the trait and state assessments were as follows for the different spiritual contexts and studies: for yoga $n_i = 11$, $\alpha_{trait/state} = 0.83/0.74$ (Studies 1 and 2); for meditation $n_i = 9$, $\alpha_{trait/state} = 0.73/0.66$ (Studies 3 and 4); for silence retreats $n_i = 13$, $\alpha_{trait/state} = 0.83/0.85$ (Studies 5 and 6); for sermon $n_i = 7$, $\alpha_{trait/state} = 0.89/0.84$ (Studies 7 and 8); and for pilgrimage $n_i = 13$, $\alpha_{trait/state} = 0.72/0.74$ (Studies 9 and 10). The items of all value scales are presented in the online supplemental material (SM3).

**Motivation.** In all studies, we assessed motivation to engage in spiritual practices in the corresponding context. The items were fully parallel in content for the trait and state assessments, and thus the number of items within spiritual contexts was identical. A sample item for the pilgrimage context is "*I am usually motivated to go on pilgrimages*" (trait assessment) and "*I am usually motivated during the pilgrimage*" (state assessment). For both the trait and state assessments, participants responded using a 5-point Likert scale, ranging from 1 (*completely disagree*) to 5 (*completely agree*).

The number of items ($n_i$) and Cronbach's alpha (correlation $r$ for the 2-item assessment, respectively) for the motivation scales in the trait and state assessments were as follows for the different spiritual contexts: for yoga $n_i = 1$ (Studies 1 and 2); for meditation $n_i = 1$ (Studies 3 and 4); for silence retreats $n_i = 4$, $\alpha_{trait/state} = 0.89/0.92$ (Studies 5 and 6); for sermon $n_i = 2$, $r_{trait/state} = 0.66/0.33$ (Studies 7 and 8); and for pilgrimage $n_i = 5$, $\alpha_{trait/state} = 0.86/0.87$ (Studies 9 and 10).

The items of the motivation scales are presented in the online supplemental material (SM4).

**Table 3 | Levels of spiritual boredom**

| Spiritual Context | Study | Trait/State | Scale/Single Item | *M* | *SD* | Skewness |
|---|---|---|---|---|---|---|
| Yoga | 1 | Trait | Scale | 2.16 | 0.66 | 0.97 |
| | 1 | Trait | Single Item | 1.85 | 0.94 | 1.08 |
| | 2 | State | Scale | 1.42 | 0.68 | 1.90 |
| | 2 | State | Single Item | 1.49 | 0.87 | 1.84 |
| Meditation | 3 | Trait | Scale | 2.39 | 0.83 | 0.24 |
| | 3 | Trait | Single Item | 2.56 | 1.05 | 0.17 |
| | 4 | State | Scale | 1.48 | 0.59 | 1.57 |
| | 4 | State | Single Item | 1.54 | 0.74 | 0.96 |
| Silence retreat | 5 | Trait | Scale | 1.63 | 0.60 | 1.23 |
| | 5 | Trait | Single Item | 1.56 | 0.82 | 1.78 |
| | 6 | State | Scale | 1.63 | 0.78 | 2.34 |
| | 6 | State | Single Item | 1.93 | 1.05 | 1.31 |
| Sermon | 7 | Trait | Scale | 3.56 | 0.94 | −0.28 |
| | 7 | Trait | Single Item | 3.60 | 1.13 | −0.40 |
| | 8 | State | Scale | 1.98 | 1.09 | 0.89 |
| | 8 | State | Single Item | 1.75 | 1.32 | 1.49 |
| Pilgrimage | 9 | Trait | Scale | 1.44 | 0.43 | 1.19 |
| | 9 | Trait | Single Item | 1.31 | 0.57 | 1.69 |
| | 10 | State | Scale | 1.35 | 0.43 | 1.55 |
| | 10 | State | Single Item | 1.24 | 0.52 | 2.05 |

All constructs were assessed by using 5-point rating scales ranging from 1 (*completely disagree*) to 5 (*completely agree*).

## Analytic strategy

**Single studies.** We used the same analytic strategy across all studies. To address our exploratory question on the levels of spiritual boredom, we report the means of the single boredom items from the spiritual boredom scales, as well as the mean values of the entire spiritual boredom scales. The reason for highlighting the values of a single item in addition to the multi-item scale is that the mean level of the single item is easier to interpret than a score aggregating answers from a multi-item scale[29]. To test our hypotheses, we calculated correlations between spiritual boredom and its assumed antecedent and outcome variables. Specifically, we examined correlations with being over- and underchallenged (antecedents), perceived value (antecedent), and motivation (effect). Prior to formal statistical hypothesis testing, we checked for violations of test assumptions. Following common recommendations[67], for correlation analyses we visually inspected scatter plots for bivariate associations. In addition, we visually inspected box plots as well as histograms and calculated the skewness of all variables (see Table 3).

**Internal meta-analyses.** To synthesize the findings across the 10 studies, we performed a series of internal meta-analyses. First, we calculated single-mean meta-analyses to estimate the average level of boredom reported across spiritual practices (i.e., the exploratory research question). We then conducted separate meta-analyses to examine the relations between boredom and over- and underchallenge (H1), value (H2), and motivation (H3). In addition, we examined whether the results of these analyses differed significantly between trait and state assessments.

As the samples of the 10 studies were independent, we included all studies in the meta-analyses. The total sample comprised *N* = 1267 participants in the single-mean meta-analyses. Due to different patterns of missing values, the meta-analytic sample sizes for the relations between boredom and overchallenge, underchallenge, value, and motivation varied between *N* = 1163 (motivation) and *N* = 1192 (value).

To estimate the mean level of spiritual boredom across the studies, we calculated a weighted mean of the means (i.e., single-mean meta-analysis, SMMA[68]). We used the raw means of the 10 boredom scales from the 10 studies. As all studies used an identical five-point Likert scale to measure boredom, the means were based on the same metric across studies. We also conducted subgroup analyses examining the mean levels of trait and state boredom separately. In addition to calculating SMMAs for the mean scores of the scales, we calculated similar SMMAs for the mean scores of the single items (i.e., each part of the scale) that assessed overall spiritual boredom. We also conducted comparable subgroup analyses for the single-item measures of spiritual boredom, distinguishing between trait and state measures. Given the variability in spiritual contexts and activities within those contexts across studies, we applied random-effects models with restricted maximum-likelihood estimation of between-study variance[69] for SMMAs of scale means and of single-item means.

To estimate the mean correlations between boredom and over-challenge, underchallenge, value, and motivation, we conducted four separate random-effects meta-analyses with restricted maximum-likelihood (REML) estimation. We used Pearson's *r* as our measure of effect size, which we transformed into Fisher's *z* for the meta-analyses. In the next step, we transformed the *z*-scores from the meta-analysis back into Pearson's *r*s[68]. In addition, we ran meta-regressions for each of the four models to examine whether the meta-analytic results varied between trait and state assessments.

### Reporting summary
Further information on research design is available in the Nature Portfolio Reporting Summary linked to this article.

## Results
### Results of single studies
**Mean levels of spiritual boredom—exploratory research question.** Table 3 shows the mean levels of spiritual boredom. Across the assessments, the means range from *M* = 1.24 (pilgrimage, state, single item) to *M* = 3.60 (sermon, trait, single item), indicating a large amount of variance across studies. The skewness of all scales and the single items are also shown in Table 3. They are negative (i.e., left-skewed) for trait sermon and positive (i.e., right-skewed) for all other contexts. This shows

**Table 4 | Antecedents and effects of spiritual boredom**

| Spiritual Context | Study | Trait/State | Overchallenge | Underchallenge | Value | Motivation |
|---|---|---|---|---|---|---|
| Yoga | 1 | Trait | 0.44*** [0.37, 0.50] | 0.54*** [0.48, 0.59] | −0.62*** [−0.67, −0.57] | −0.49*** [−0.55, −0.43] |
| | 2 | State | 0.29* (0.030) [0.16, 0.41] | 0.72*** [0.65, 0.78] | −0.56*** [−0.64, −0.46] | −0.66*** [−0.73, −0.57] |
| Meditation | 3 | Trait | 0.36** (0.004) [0.24, 0.47] | 0.48*** [0.37, 0.58] | −0.56*** [−0.64, −0.47] | −0.36+ (0.060) [−0.52, −0.17] |
| | 4 | State | 0.35** (0.006) [0.23, 0.46] | 0.42*** [0.30, 0.52] | −0.28* (0.029) [−0.40, −0.16] | −0.28* (0.026) [−0.40, −0.16] |
| Silence retreat | 5 | Trait | 0.45*** [0.35, 0.53] | 0.29** (0.009) [0.19, 0.40] | −0.53*** [−0.61, −0.45] | −0.47*** [−0.55, −0.38] |
| | 6 | State | 0.55*** [0.43, 0.66] | 0.38* (0.018) [0.22, 0.51] | −0.61*** [−0.70, −0.50] | −0.58*** [−0.68, −0.46] |
| Sermon | 7 | Trait | 0.03 (0.547) [−0.02, 0.09] | 0.23*** [0.18, 0.28] | −0.68*** [−0.71, −0.65] | −0.58*** [−0.61, −0.54] |
| | 8 | State | 0.75*** [0.70, 0.79] | 0.56*** [0.49, 0.63] | −0.39*** [−0.47, −0.30] | −0.23* (0.023) [−0.33, −0.13] |
| Pilgrimage | 9 | Trait | 0.53*** [0.47, 0.59] | 0.41*** [0.34, 0.47] | −0.59*** [−0.64, −0.53] | −0.53*** [−0.58, −0.47] |
| | 10 | State | 0.50*** [0.43, 0.56] | 0.34*** [0.25, 0.42] | −0.37*** [−0.45, −0.29] | −0.33*** [−0.41, −0.25] |

All constructs were assessed by using 5-point rating scales ranging from 1 (*completely disagree*) to 5 (*completely agree*). For *p*-values > 0.001, the exact *p*-values (two-sided test) are provided in parentheses. 95% confidence intervals are presented.
+) $p < 0.10$; *) $p < 0.05$; **) $p < 0.01$; ***) $p < 0.001$.

that there are few relatively low scores for trait sermon, while there are few relatively high scores for the other assessments[70].

For trait sermon, we found that 69.83% of the ratings were 4 (or 5 on the 5-point Likert scale; for state sermon, this was 54.78%. Across all scales/items, the mean percentage of 4- and 5-point responses was 12.63 (for all percentages on scores of 3–5, 4–5 and 5 see online supplemental material SM5).

**Antecedents and effects of spiritual boredom—H1, H2, H3.** Table 4 presents results on the antecedents and effects of spiritual boredom. Means, standard deviations as well as intercorrelations of all scales within the 10 studies are presented in the online supplemental material SM6.

Supporting Hypothesis 1, spiritual boredom was significantly positively correlated with being overchallenged in all 10 studies (all *p*s < 0.05, see Table 4); the zero correlation for state sermon boredom was an exception. Spiritual boredom was also significantly positively correlated with being underchallenged in all studies (all *p*s < 0.05, see Table 4). Significant correlations across studies ranged from $r = 0.29$ to $r = 0.75$ for being overchallenged and from $r = 0.23$ to $r = 0.72$ for being underchallenged.

In line with Hypothesis 2, spiritual boredom was significantly negatively correlated (all *p*s < 0.05, see Table 4) with perceived value in all studies, with correlations ranging from $r = -0.28$ to $r = -0.68$. Our results therefore fully support Hypothesis 2.

Regarding Hypothesis 3, spiritual boredom was significantly negatively correlated with motivation in all studies (one correlation *p* < 0.10, all other *p*s < 0.05, see Table 4), with correlations ranging from $r = -0.28$ to $r = -0.66$. Our results therefore clearly support Hypothesis 3.

While supporting the study hypotheses, the results also imply that there was a large heterogeneity in the strength of the relations between spiritual boredom and its theorized antecedents and effects across spiritual contexts, and across trait and state assessments.

**Results of the internal meta-analyses**
**Meta-analysis of mean boredom—exploratory research question.** To estimate the mean level of spiritual boredom across the five spiritual contexts, we conducted SMMAs for the mean values of the boredom scales used in the respective studies, as well as for the means of the single items assessing overall boredom. Synthesizing the scale means from all 10 studies (i.e., five spiritual contexts, each with one trait and one state-related study), our random-effects model yielded a statistically significant mean boredom value of $\bar{M} = 1.91$ (95% CI [1.48, 2.33]). This indicates a relatively low aggregated mean level of boredom (on a response scale from 1 [*strongly disagree*] to 5 [*strongly agree*]).

The between-study heterogeneity was significant and high ($Q = 1821.35$, $p < 0.001$, $I^2 > 99.99$), indicating considerable variability in boredom levels across the different studies[71]. Participants reported the highest levels of boredom in Study 7 (sermon—trait; $M = 3.56$, $SD = 0.94$, see Table 3) and the lowest levels in Study 10 (pilgrimage—state; $M = 1.35$, $SD = 0.43$, see Table 3). Figure 2 shows the distributions for the single items assessing overall boredom and for the multi-item scales. When analysing trait and state assessments separately, individuals reported higher mean boredom in trait assessments compared to state assessments, although the confidence intervals overlapped (see online supplemental material SM7 for detailed results).

We observed a very similar pattern when we repeated the SMMA for the single-item boredom measures from each study ($\bar{M} = 1.88$, 95% CI [1.44, 2.33], $Q = 1371.79$, $p < 0.001$, $I^2 > 99.99$). Again, the highest levels of boredom were reported in Study 7 (sermon—trait; $M = 3.60$, $SD = 1.13$, see Table 3), and the lowest levels were reported in Study 10 (pilgrimage—state; $M = 1.24$, $SD = 0.52$, see Table 3). Consistent with the SMMA of the scale means, trait boredom was generally more pronounced than state boredom, although the confidence intervals overlapped (for detailed results, see online supplemental material SM7).

**Meta-analyses of antecedents and outcomes of spiritual boredom —H1, H2, H3.** To estimate the overall effect sizes of the relations between spiritual boredom and overchallenge, underchallenge, value, and motivation, we conducted a series of four meta-analyses (Table 5; see Fig. 3 for a graphical illustration of the findings). The random-effects models yielded significant, positive effect sizes for overchallenge ($\bar{r} = 0.44$, 95% CI [0.30, 0.56]) and underchallenge ($\bar{r} = 0.44$, 95% CI [0.34, 0.53], see Table 5), indicating that being over- or underchallenged is substantially positively associated with experiencing boredom in different spiritual contexts. Conversely, spiritual boredom was significantly negatively associated with value ($\bar{r} = -0.54$, 95% CI [−0.61, −0.45], see Table 5), suggesting that lower subjective value of a spiritual practice is meaningfully associated with higher levels of spiritual boredom. Similarly, boredom was significantly negatively

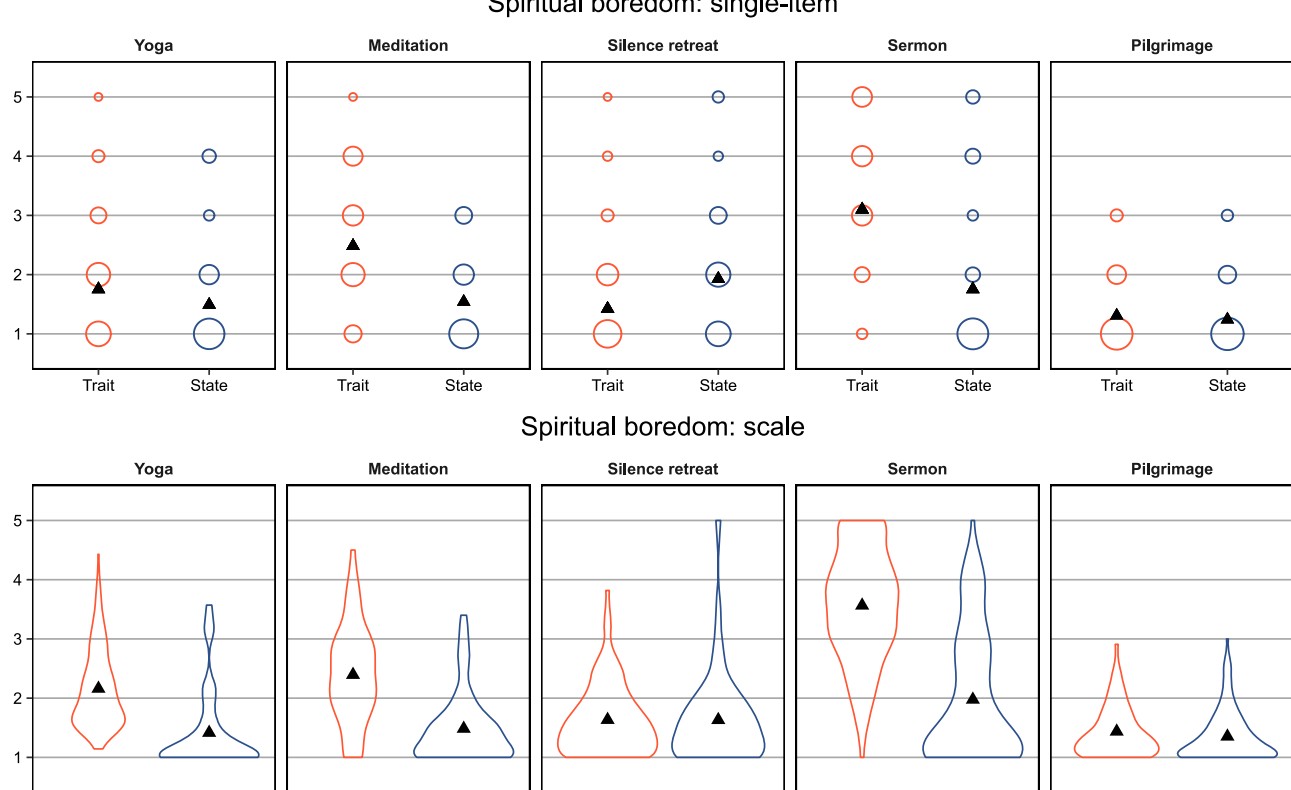

**Fig. 2 | Descriptive statistics for single-item and scale indicators of spiritual boredom.** *Note.* Top panel refers to levels of spiritual boredom as assessed via the single-items assessing spiritual boredom in the respective context as a whole: "When I practice yoga, I usually get bored" (yoga – trait); "During the yoga session I had just completed I was bored" (yoga – state); "When I practice meditation, I usually get bored" (meditation – trait); "During the meditation session I had just completed I was bored" (meditation - state); "During silence retreats, I usually get bored" (silence retreat – trait); "In the current or completed silence retreat I was bored" (silence retreat – state); "I generally find sermons boring" (sermon – trait); "I found the sermon boring today" (sermon – state); "When I'm on a pilgrimage, I usually get bored" (pilgrimage – trait); "Please think about the last days of your pilgrimage. …I felt bored" (pilgrimage – state). The relative size of the circles corresponds to the relative frequencies of the rating categories within studies. The bottom panel shows the distributions of participant's scores on the spiritual boredom multi-items scales used in the respective studies. The y-axes of both panels depict the five-point Likert response format that was used to assess boredom in all studies (i.e., a 5-point Likert scale ranging from 1 [*completely disagree*] to 5 [*completely agree*]). The black triangles indicate the mean values in both panels. The orange circles/lines refer to the trait assessment, the blue circles/lines refer to the state assessment. Sample sizes (trait/state): yoga $n$ = 159/57 participants; meditation $n$ = 63/61 participants; silence retreat $n$ = 90/40 participants; sermon $n$ = 414/97 participants; pilgrimage $n$ = 162/124 participants.

**Table 5 | Meta-analyses of correlations with spiritual boredom**

| Model | Mean $r$ | $k$ | $I^2$ | $Q$ |
|---|---|---|---|---|
| Overchallenge | 0.44*** [0.30, 0.56] | 10 | 85.79 | 90.75*** |
| Underchallenge | 0.44*** [0.34, 0.53] | 10 | 73.34 | 36.24*** |
| Value | −0.54*** [−0.61, −0.45] | 10 | 70.45 | 34.49*** |
| Motivation | −0.46*** [−0.55, −0.37] | 10 | 66.88 | 26.74** |

In brackets, 95 percent confidence intervals are provided. $k$ indicates the number of included studies. $I^2$ denotes the percentage of variance explained by between-study heterogeneity relative to sampling variance. $Q$ is a parameter used to quantify between-study heterogeneity[68].
**$p < 0.01$. ***$p < 0.001$.

associated with motivation to engage in the spiritual practice ($\bar{r} = -0.46$, 95% CI [−0.55, −0.37], see Table 5).

The proportion of variance explained by between-study heterogeneity was high for overchallenge ($Q = 90.75$, $p < 0.001$, $I^2 = 85.79$) and underchallenge ($Q = 36.24$, $p < 0.001$, $I^2 = 73.34$), and moderate for value ($Q = 34.49$, $p < 0.001$, $I^2 = 70.45$) and motivation ($Q = 26.74$, $p < 0.001$, $I^2 = 66.88$). This indicates that the relations between boredom and over- and underchallenge varied more substantially across the 10 studies compared to the relations between boredom, on the one hand, and value and motivation, on the other[71].

To examine whether the results varied between trait and state assessments, we ran meta-regressions with trait versus state assessment as the predictor variable. The results were not significantly impacted for overchallenge, underchallenge, and motivation. However, trait versus state significantly moderated the effect sizes for value ($\beta = 0.27$, 95% CI [0.12, 0.41], see online supplemental material SM7). This indicates that the correlations between spiritual boredom and value were significantly stronger for trait assessments ($\bar{r} = -0.62$, 95% CI [−0.67, −0.56]) than for state assessments ($\bar{r} = -0.42$, 95% CI [−0.51, −0.31], see online supplemental material SM7).

## Discussion

In this research, we focused on spiritual boredom, a topic that has been largely neglected in previous empirical research. Our aim was to gain insight

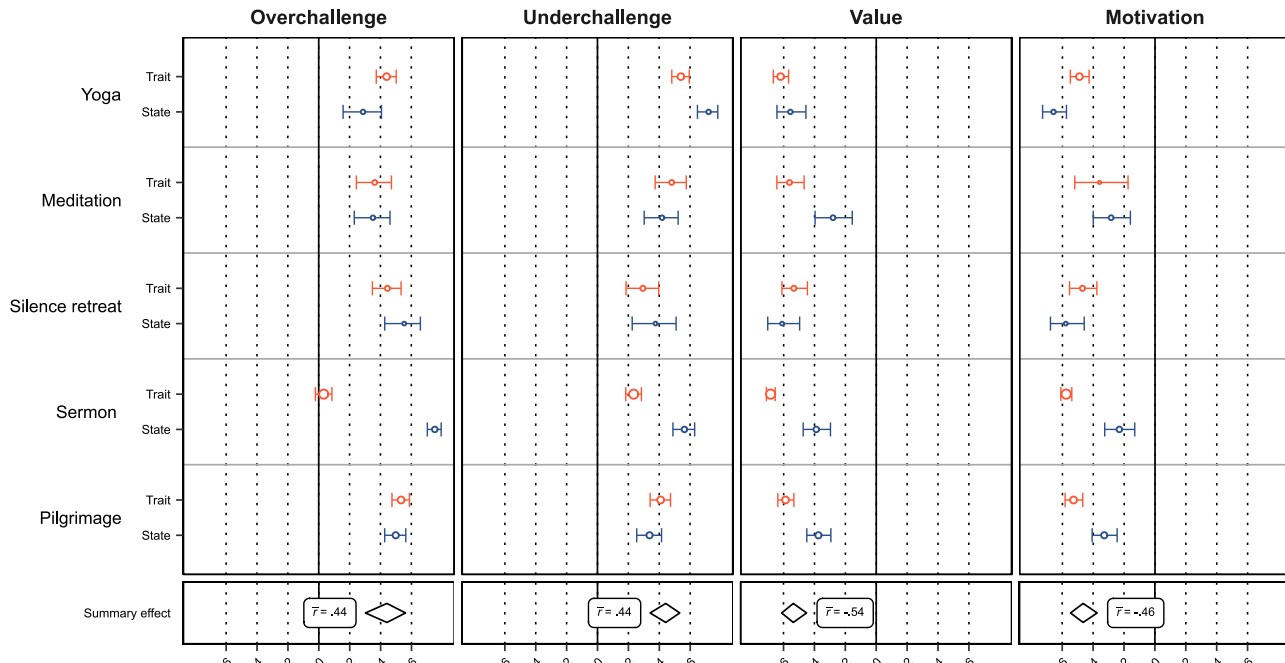

**Fig. 3 | Forest plot for meta-analyses of correlations with spiritual boredom.** *Note*. The x-axis of the panels indicates summary *rs*. The dotted vertical lines correspond to well-established effect size benchmarks[97], whereas the solid black line denotes *r* = 0.00. The y-axis refers to the different spiritual activities, each grouped by trait and state assessments. The four horizontally arranged panels display the results separately for overchallenge, underchallenge, value, and motivation. The circles with horizontal bars denote effect sizes and their confidence intervals, with sizes of the circles indicating the relative sample sizes of the studies. The boxes below the main panels contain meta-analytic summary effects. The width of the diamonds corresponds to their confidence intervals. The orange lines refer to the trait assessment, the blue lines refer to the state assessment.

into the levels of boredom experienced during spiritual practice (an exploratory research question) and to examine its antecedents and effects based on Pekrun's[9,10,12] control-value theory (CVT). Specifically, we examined whether inadequate levels of perceived control (i.e., being over-challenged or underchallenged) were associated with high levels of spiritual boredom (Hypothesis 1) and whether perceived value was associated with low levels of spiritual boredom (Hypothesis 2). In terms of the impact of spiritual boredom, we focused on one core variable: motivation to engage in spiritual practice, which we hypothesized would be negatively related to spiritual boredom (Hypothesis 3). As prototypical examples, we examined five spiritual contexts: yoga, meditation, silence retreats, sermons (as part of Catholic services), and pilgrimage. In each context, we conducted two independent studies—one focusing on trait-related assessments and the other on state-related assessments. We synthesized the results of the 10 studies using meta-analyses.

## Occurrence of spiritual boredom

There are scattered indicators from quantitative studies regarding the occurrence of spiritual boredom, but little empirical evidence on its levels. In a previous study[7], the reported level of "void" during mindfulness meditation ranged from 2.4 to 2.8 on a 7-point Likert scale, with 1 indicating "*completely disagree*" and 7 "*completely agree*." Our meta-analysis revealed a mean level of spiritual boredom of $\bar{M}$ = 1.91 for the multi-item scales and $\bar{M}$ = 1.88 for the single items (assessed on a 5-point Likert scale ranging from 1 [*completely disagree*] to 5 [*completely agree*]). When considering the differences in the metric used, these findings are similar to prior results[7]. Both sets of results reflect mean levels of approximately 20 to 25% of the maximum scale value.

Our internal meta-analysis showed that trait boredom was generally more pronounced than state boredom. One reason may be that not all participants in the trait assessments were currently engaged in spiritual practice (even though they all had experience with it), in contrast to the state statements where all participants were currently practicing. For some participants in the trait assessments, high levels of boredom may actually have been a

reason for not currently practicing, leading to higher reports of trait spiritual boredom among these currently non-practicing participants, and consequently to higher overall mean levels of trait spiritual boredom. Furthermore, our findings are in line with many other studies showing higher levels of trait compared to state emotion scores. A major general reason for higher trait scores, as outlined in the literature, might be peak effects, that is, the over-weighting of very intense experiences in retrospective judgments[72–74].

Compared to the other spiritual contexts, the mean value of trait sermon boredom was relatively high (*M* = 3.56, *SD* = 0.94 for the sum scale; *M* = 3.60, *SD* = 1.13 for the single boredom item; rated on a scale from 1 [*completely disagree*] to 5 [*completely agree*]). Many participants may attend Catholic services for reasons unrelated to the sermon, such as enjoying the singing of hymns or the quiet elements of the service. Consequently, some attendees may tolerate the sermon, even if boring, in order to experience the other aspects of services they enjoy.

We found that the level of boredom during pilgrimages was rather low. This may be due to the inherent variability of the pilgrimage experience. Elements such as varied landscapes, changing weather, manageable challenges and encounters with a wide range of people are likely to contribute to this variability and help to perceive value and reduce feelings of boredom.

Except for sermon boredom, the mean levels of spiritual boredom found in our research were below the midpoint of the response scale (i.e., a value of 3.00). However, we observed instances of boredom scores above the midpoint of the scale in all contexts except pilgrimage and the state single item in the context of meditation. This supports the view that boredom is indeed a non-negligible emotion in spiritual practices. For those who wish to interpret the mean levels of spiritual boredom within specific spiritual contexts in relation to participants' engagement in spiritual practices, we provide descriptive statistics on participants' current spiritual practices across all 10 studies in the online supplementary material (SM8).

In summary, our studies show that there is a relatively low but significant mean level of spiritual boredom. Although qualitative studies, anecdotes, and artworks (e.g., individuals sleeping during sermons) have

suggested the presence of significant levels of spiritual boredom, our study quantitatively demonstrates its prevalence.

## Antecedents of spiritual boredom

**Nonoptimal levels of control—being over- or underchallenged.** Consistent with assumptions derived from CVT[9,10,12] and in line with previous research in other contexts (e.g., academic boredom[29,48,75]), we found that spiritual boredom was positively related to non-optimal levels of control. Our internal meta-analyses revealed that spiritual boredom was negatively related to both being overchallenged ($\bar{r} = 0.44$) and underchallenged ($\bar{r} = 0.44$). Spiritual boredom was significantly positively correlated with over- and underchallenge in all 10 studies, except for the non-significant relation between boredom and overchallenge in the sermon-related trait assessment. A possible reason for this unexpected finding could be that overchallenge was assessed with a single item in this study (in contrast to the other studies). The validity of this item (i.e., "*The content of a sermon usually overchallenges me*") may be compromised because it might have been interpreted in different ways. For example, overchallenge could be interpreted as relating to the content of the sermon, its length, or the complexity of its language. Future studies might benefit from using multi-item scales to more accurately assess overchallenge during sermons.

The results of the meta-analysis showed that the relations between boredom and over- and underchallenge showed more heterogeneity across the 10 studies than the relations between boredom and value and between boredom and motivation. One explanation may be that spiritual boredom might differ in its sensitivity to non-optimal levels of control depending on the spiritual context. For example, being overchallenged while listening to a sermon might lead to greater boredom than being overchallenged while practicing yoga, because the level of challenge in listening to a sermon might be subjectively perceived to be less controllable than in individual yoga practice.

**Perceived value.** According to most theories on boredom, including CVT, perceived value reduces boredom[3,24]. Consistent with numerous studies in other contexts, we found significant negative relations between perceived value and spiritual boredom in all 10 studies. Our meta-analytic results revealed a mean negative relation of $\bar{r} = -0.54$. Any spiritual practice that is practiced regularly can easily become an unthinking routine whose value is not sufficiently appreciated. This reduced value can lead to spiritual boredom, which, in turn, can further reduce the value of the spiritual practice, creating a negative downward spiral. While previous research has examined whether individuals who perceive more meaning in life experience less boredom[15], our study examined the relation between the perceived value of the spiritual practice itself and the boredom experienced during that practice. The findings suggest that perceived value can play a crucial role in the experience of spiritual boredom.

Our meta-analytic findings show that the correlations between spiritual boredom and value were significantly stronger for trait assessments than for state assessments. One explanation could be that it is intuitively plausible that low value leads to boredom; this subjective belief may have influenced the responses in the trait assessment. Trait self-reports are known to be more sensitive to subjective beliefs than state assessments[76,77]. Nevertheless, the correlations were substantially negative in both types of assessments (i.e., $\bar{r} = -0.62$ for the trait and $\bar{r} = -0.43$ for the state assessment).

## Effects of spiritual boredom—motivation to engage

Consistent with CVT, we found significant negative relations between spiritual boredom and motivation for spiritual practice in all 10 studies. Our meta-analytic results showed a mean negative relation of $\bar{r} = -0.46$. These findings are comparable to a prior study[7] reporting a significant negative relation ($r = -0.54$) between "void" and motivation during guided mindfulness meditation. However, our study focused directly on spiritual boredom. The findings suggest that high levels of boredom during spiritual practice strongly reduce motivation for further practice.

In sum, with respect to our exploratory research question, our findings reveal significant and thus non-negligible levels of spiritual boredom.

Furthermore, our findings clearly support our three explanatory hypotheses: that spiritual boredom is positively related to non-optimal levels of control (i.e., being over- or underchallenged; Hypothesis 1), negatively related to value (Hypothesis 2), and negatively related to motivation (Hypothesis 3).

## Limitations

Some limitations of the present study should be noted and can inform directions for future research. First, we relied on self-report data to assess spiritual boredom, antecedents, and effects. As such, the findings may have been influenced by response sets that can impact self-report[78]. To reduce potential biases, future studies of spiritual boredom could also include objective assessments of spiritual boredom components, such as physiological measures of reduced arousal[38,79].

Second, while we focused on control and value as antecedents of boredom, recent models of boredom have pointed to the important role that attentional failures play in characterizing or causing boredom[14,80,81]. It would be important for future work to examine the role of attention in spiritual boredom.

Third, our study focused on one specific effect of spiritual boredom: its impact on motivation to engage in spiritual practices. Future research could extend the scope by exploring additional effects of spiritual boredom, such as its influence on the frequency and duration of spiritual practices, as well as its impact on self-regulation during these practices (e.g., during meditation).

Fourthly, we recruited participants for our study using snowball sampling, aiming to encourage as many individuals as possible to take part. Future studies on spiritual boredom could perform power analyses to ensure adequate sample sizes. Our initial findings on spiritual boredom may serve as a helpful foundation for such analyses.

In addition, it is important to note that our approach does not allow to draw conclusions about the causal ordering of the variables. Future studies should use experimental and longitudinal designs to address this limitation. Also, as noted earlier, our samples were not fully comparable between the trait and state assessments. The trait samples included participants who may be infrequently engaging in spiritual practices, whereas the state assessments may have included individuals who practiced more frequently. All participants in the state assessments were currently practicing; this was not the case for the trait sample. Future studies could use the same participants to examine both traits and state spiritual boredom within one sample.

Finally, our study focused on Western and German-speaking samples and five exemplary domains of spiritual boredom. Future research could expand this scope to test the generalizability of the present findings across cultures and additional domains, such as spiritual dance, mindfulness practices, spiritual breathing, and other religious practices.

## Implications for research and practice

An implication of our study is that spiritual boredom, in terms of its levels and potential effects, is a critically important emotion to be considered in future research and practice. As such, our hope is that the initial theoretical framework and empirical findings of the present set of studies will stimulate further research into spiritual boredom.

Regarding antecedents of spiritual boredom, our findings are consistent with previous studies of boredom in other contexts (e.g., academic contexts[29,48,75]), which suggest that suboptimal levels of control (i.e., over- or underchallenge) increase spiritual boredom. As discussed previously[7], many spiritual practices are typically not individualized (e.g., guided meditations, yoga practices, sermons, silent retreats, prayers, chanting), which can often lead to experiences of over- or underchallenge. While there is an ongoing debate in research on education and the workplace about how practices can be personalized to improve wellbeing and growth, this discussion is largely absent for spiritual contexts. It may be important to discuss personalization for spiritual practices as well.

Our research indicates that individualized spiritual practices could potentially help alleviate spiritual boredom. For example, meditation groups could be divided into smaller groups based on individual preferences, such as guided versus unguided meditation, different lengths of meditation

sessions, or meditation with and without music. For sermons, alternatives could be offered for those who do not find sermons engaging, such as books with inspiring images or podcasts with spiritual poetry or thought-provoking questions on spiritual topics. While such materials are sometimes offered to children during worship, they are less commonly used for adults. Beyond these specific examples, principles of individualization applied in other contexts (e.g., education) could be adapted to develop more perso-nalized spiritual practices. Future research could focus on designing and evaluating programmes that incorporate these individualized approaches to assess their impact on spiritual boredom.

Our research is consistent with studies of boredom in other contexts (e.g., education, work) in suggesting that emphasizing the value of practices can help reduce boredom. For example, highlighting the importance of these practices in courses could be beneficial. Various meta-analyses have shown positive associations between spirituality and mental health[82–84], physical health[85–87], well-being[88], social participation[89], and social responsibility[90]. Furthermore, spirituality has been negatively associated with physical and sexual aggression[91] and delinquent behavior[92]. However, when outlining the benefits or value of specific spiritual practices, it is crucial to base such statements on empirical research findings.

In terms of the consequences of boredom, our research suggests that spiritual boredom can reduce motivation for spiritual practice. As such, spiritual boredom can be seen as a previously unrecognized indicator or facet of spiritual struggle[59,60]. Course leaders should attend to signs of boredom among participants and respond appropriately, for example by providing breaks. In addition, it might be valuable to address spiritual boredom as a topic for discussion. This could encourage a stimulating exchange on the topic, as boredom is often seen as an opportunity for self-reflection, but paradoxically seems to reduce the motivation to engage in spiritual practice in the first place[32].

Future research could explore how spiritual boredom affects such motivation to reflect on the spiritual practice. Future studies could also investigate the impact of boredom on individuals' motivation to apply spiritual insights to daily life interactions. In addition, as mentioned earlier, examining the frequency and duration of spiritual practices, as well as self-regulation skills during these practices (e.g., during meditation), could provide further insight into the consequences of spiritual boredom.

Our findings indicate that the CVT, originally developed in the context of achievement emotions, serves as an appropriate theoretical framework for explaining the antecedents and effects of spiritual boredom. Consistent with ongoing theoretical developments suggesting that CVT is applicable beyond academic contexts[12], our findings encourage the use of CVT for further research on spiritual boredom (e.g., exploring additional effects of spiritual boredom, such as its impact on the quality of spiritual practice).

In this research, we developed a total of 10 boredom scales, including one trait and one state spiritual boredom scale each for the spiritual contexts of yoga (YBS-T, YBS-S), meditation (MBS-T, MBS-S), silence retreat (SRBS-T, SRBS-S), sermon (SBS-T, SBS-S), and pilgrimage (PBS-T, PBS-S; see online supplemental material SM1 for the wording of the items of all scales). Our initial results indicate high convergent validity of the newly developed scale, as evidenced by the strong correlations between the MSBS-SF (i.e., the short form of the Multidimensional State Boredom Scale[66]) and the YBS-S, as well as the MSBS-SF and the MBS-S. Future research on spiritual boredom could use these scales or adapted versions (e.g., short versions or versions that refer to other spiritual contexts). Since all these scales include a single item assessing the overall level of boredom in each spiritual context, these single items can also be used when the number of items to be assessed is limited (e.g., due to study design, as in experience sampling or laboratory studies).

## Conclusion

In the face of current global crises, such as the climate crisis and ongoing wars, people may seek out spiritual growth and practices in pursuit of social connectedness and empathy, possibly countering tendencies towards ego-centrism and blind competition. This shift can encourage behavior that benefits the common good in our society.

By addressing and reducing spiritual boredom, our findings could help promote engagement in spiritual practices, ultimately supporting personal and collective spiritual growth. Spiritual boredom could be alleviated by designing spiritual practices in a way that reduces over- or underchallenge (e.g., through individualized practices) and by increasing the perceived value of the practices (e.g., by emphasizing the importance of the practices for daily life).

## Data availability

All data, on which this paper is based, are available at https://doi.org/10.17605/OSF.IO/G5PCU.

## Code availability

All statistical analyses were conducted in R 4.4.2[93]. The R package *meta* was used to calculate the SMMAs[94]. For meta-analyses of effect sizes, we used the *metafor* package[95]. For data visualization, we used the *ggplot2* package[96]. The code allowing to reproduce the presented analyses is available on the Open Science Framework at https://doi.org/10.17605/OSF.IO/G5PCU.

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

## Author contributions

T.G.: Conceptualization, Formal analysis, Writing—original draft. J.F.: Formal analysis, Methodology, Software, Visualization, Writing—original draft. L.S.: Writing—original draft. L.K.: Formal analysis, Writing—original draft. S.S.: Writing—original draft. L.B.: Conceptualization, Methodology, Investigation. Y.L.D.: Conceptualization, Methodology, Investigation. C.P.: Conceptualization, Methodology, Investigation. B.S.: Conceptualization, Methodology, Investigation. S.W.: Conceptualization, Methodology, Investigation. W.A.P.vanT.: Writing—original draft. R.P.: Writing—original draft. All authors: Writing—review & editing.

## Competing interests

The authors declare no competing interests.
