## [Transparent Peer Review file · Communications Psychology]

Spiritual boredom: First evidence for associations with over- and underchallenge, lack of value, and disengagement

Corresponding Author: Dr Thomas Goetz

Version 0:

Decision Letter: First round

Decision Letter:

Dear Dr Goetz,

Thank you for your patience during the peer-review process. Your manuscript titled "Spiritual Boredom: Exploring the Occurrence, Antecedents, and Effects of a Neglected Emotion" has now been seen by 2 reviewers, and I include their comments at the end of this message. They find your work of interest but raised some important points. We are interested in the possibility of publishing your study in Communications Psychology, but would like to consider your responses to these concerns and assess a revised manuscript before we make a final decision on publication.

We therefore invite you to revise and resubmit your manuscript, along with a point-by-point response to the reviewers. Please highlight all changes in the manuscript text file.

Editorially, we consider it important that the revised manuscript clarifies definitions/concepts and highlights the exploratory character of the work. Please address all methodological questions and concerns raised by Reviewer 2. Any concerns that cannot be addressed should be discussed as limitations.

I am attaching an Editorial Requests Table that details critical reporting requirements for the revised manuscript. Please attend to each item and ensure your manuscript is fully compliant. If your revised manuscript is not aligned with these requests on major issues, such as those concerning statistics, it may be returned to you for further revisions without re-review.

Please submit the following items:

- Revised manuscript
- Point-by-point response to the referees' comments
- Cover letter (as a separate document)
- [Nature Research Reporting Summary](https://www.nature.com/documents/nr-reporting-summary.zip)
- [Editorial Policy Checklist](https://www.nature.com/documents/nr-editorial-policy-checklist.pdf)
- Completed Editorial Request Table (attached).

via this link: Link Redacted .

Additional guidance is available in our style and formatting guide Communications Psychology formatting guide.

Best regards,

Jennifer Bellingtier

Jennifer Bellingtier, PhD
Senior Editor
Communications Psychology

REVIEWER EXPERTISE:
Reviewer #1 religion/spirituality
Reviewer #2 boredom

REVIEWER REPORTS:

Reviewer #1 (Remarks to the Author):

This manuscript is excellent in numerous respects, demonstrating a high level of scholarly rigor and creativity. The authors address a novel and important topic that is likely to contribute meaningfully to the field. The following aspects of the manuscript stand out as particularly noteworthy: It addresses a novel topic, provides a comprehensive literature review, is written in an engaging and clear style, gives a compelling theoretical rationale, includes large sample sizes and diverse contexts, presents rigorous analyses and thorough interpretation of results.

While the manuscript is exceptionally well-crafted, I would recommend that the authors engage with the literature on religious and spiritual struggles, which is a well-established area of study. Notably, Julie Exline has written several comprehensive reviews on this topic, which could provide valuable insights for the manuscript. This body of literature examines the challenging and potentially negative aspects of spirituality, making it particularly relevant to the study of spiritual boredom. The notion of boredom potentially as an unrecognized form of spiritual struggle is intriguing and aligns closely with themes explored in this literature. Incorporating these perspectives could enrich the manuscript's theoretical framework and deepen its practical implications.

Reviewer #2 (Remarks to the Author):

The present manuscript offers valuable insights into the topic of boredom in spiritual settings, which, as the authors have rightly highlighted, is a much neglected yet important area of research. The authors examined whether spiritual boredom is associated with perceived control, perceived values, and motivation to engage. They conducted 10 correlational studies across five different spiritual contexts and provided an internal meta-analysis. I appreciate the authors' efforts to address a critical gap in the literature through a comprehensive approach. I have several suggestions that I hope will help further strengthen the manuscript:

1. The authors are encouraged to provide stronger justifications for why it is necessary to single out and study "spiritual boredom" as a distinct type of boredom. The manuscript theorizes that spiritual boredom aligns with the general features of boredom and academic boredom, and the findings support this. However, if spiritual boredom represents the same feeling of boredom manifesting in a different context, further discussion on why studying it as a separate construct is warranted would be helpful. For instance, studying "gaming boredom" or "commuting boredom" would likely yield similar results. While I agree

with the authors that it is important to study spiritual boredom, more effort is needed to highlight its unique significance so that readers can fully appreciate its value. For example, academic boredom warrants specific attention because it impacts a distinct population (students), occurs in a more structured and restricted setting (schools), and has unique consequences (e.g., affecting academic performance and learning motivation). Could the authors elaborate on whether spiritual boredom has similarly distinctive characteristics, affects a specific population, and/or carries unique implications?

2. Relatedly, further clarification on how spiritual boredom differs from boredom in other contexts, such as academic boredom or boredom in general, would be valuable.

3. Given the manuscript's focus on the Control-Value Theory, elaborating on why this framework—originally developed to explain emotions in academic and achievement settings—serves as an appropriate lens for understanding boredom in spiritual contexts would strengthen the discussion.

4. While acknowledging that research on boredom in spiritual contexts is limited, it would be helpful to include a review of relevant literature on how constructs like meaning, perceived control and motivation the engage relate to spiritual experiences.

5. Please provide more methodological details for the studies. For example:

- What exclusion criteria were applied across the studies?
- Were power analyses conducted to ensure adequate sample sizes?
- What information was provided to participants about the studies during recruitment?
- Did the authors collect information about participants' engagement in spiritual practices, such as the frequency or duration of their practice or their years of experience? Providing such details would offer important context for interpreting the findings.
- Please provide more information on the scale development process, as well as the scales' reliability and validity.

6. Regarding the state measures, were participants surveyed immediately after engaging in spiritual practices? Also, were all participants currently practicing spiritual practices? If so, this could be a limitation of the research, as those who find spiritual practices particularly boring may have already quit and were not included in the study.

I hope these suggestions are helpful in further refining the manuscript and strengthening its contribution to the field.

Version 1:

Decision Letter: Second round

Decision Letter:

Dear Dr Goetz,

Your manuscript titled "Spiritual Boredom: Exploring the Occurrence, Antecedents, and Effects of a Neglected Emotion" has now been seen by our reviewers, whose comments appear below. In light of their advice I am delighted to say that we are happy, in principle, to publish a suitably revised version in Communications Psychology.

We therefore invite you to revise your paper one last time to address the remaining concerns of our reviewers and a list of editorial requests. At the same time we ask that you edit your manuscript to comply with our format requirements and to maximise the accessibility and therefore the impact of your work.

EDITORIAL REQUESTS:

SUBMISSION INFORMATION:

OPEN ACCESS:

* DATA AVAILABILITY:

Link Redacted

Best regards,

Jennifer Bellingtier

Jennifer Bellingtier, PhD
Senior Editor
Communications Psychology

REVIEWERS' EXPERTISE:

Reviewer #1 religion/spirituality
Reviewer #2 boredom

REVIEWERS' COMMENTS:

Reviewer #1 (Remarks to the Author):

Thanks to the authors for responding to my previous comments. I believe the manuscript will make a valuable contribution to the literature.

Reviewer #2 : (no further comments)

Author response: First round

Author Response to Reviewer Comments

Reviewer 1

This manuscript is excellent in numerous respects, demonstrating a high level of scholarly rigor and creativity. The authors address a novel and important topic that is likely to contribute meaningfully to the field. The following aspects of the manuscript stand out as particularly noteworthy: It addresses a novel topic, provides a comprehensive literature review, is written in an engaging and clear style, gives a compelling theoretical rationale, includes large sample sizes and diverse contexts, presents rigorous analyses and thorough interpretation of results.

While the manuscript is exceptionally well-crafted, I would recommend that the authors engage with the literature on religious and spiritual struggles, which is a well-established area of study. Notably, Julie Exline has written several comprehensive reviews on this topic, which could provide valuable insights for the manuscript. This body of literature examines the challenging and potentially negative aspects of spirituality, making it particularly relevant to the study of spiritual boredom. The notion of boredom potentially as an unrecognized form of spiritual struggle is intriguing and aligns closely with themes explored in this literature. Incorporating these perspectives could enrich the manuscript's theoretical framework and deepen its practical implications.

Response. Thank you for your positive evaluation of our work. Following your very helpful suggestion, we now refer to the research on religious and spiritual struggles in the sections “Spiritual Boredom / Effects” and “Implications for Research and Practice.” We outline that spiritual boredom might be seen as an up to now unrecognized indicator or form of spiritual struggle. We think that referring to research on spiritual struggle further embeds our research into the broader field of research on spirituality.

[Spiritual Boredom / Effects]

Thus, spiritual boredom may represent a previously unrecognized indicator or form of spiritual struggle, with research on the topic emphasizing experiences of tension, strain, or conflict in relation to religion and spirituality (Exline et al. 2014; Pargament & Exline 2022). (p. 12)

[Implications for Research and Practice]

In terms of the consequences of boredom, our research suggests that spiritual boredom can reduce motivation for spiritual practice. As such, spiritual boredom can be seen as a previously unrecognized indicator or facet of spiritual struggle (Exline et al., 2014; Pargament & Exline, 2022). Course leaders should attend to signs of boredom among participants and respond appropriately, for example by providing breaks. In addition, it might be valuable to address spiritual boredom as a topic for discussion. (p. 34)

Exline, J. J., Pargament, K. I., Grubbs, J. B., & Yali, A. M. (2014). The Religious and Spiritual Struggles Scale: Development and initial validation. *Psychology of Religion and Spirituality*, 6(3), 208–222. <https://doi.org/10.1037/a0036465>

Pargament, K. I., & Exline, J. J. (2022). *Working with spiritual struggles in psychotherapy: From research to practice*. Guilford.

Reviewer 2

The present manuscript offers valuable insights into the topic of boredom in spiritual settings, which, as the authors have rightly highlighted, is a much neglected yet important area of research. The authors examined whether spiritual boredom is associated with perceived control, perceived values, and motivation to engage. They conducted 10 correlational studies across five different spiritual contexts and provided an internal meta-analysis. I appreciate the authors' efforts to address a critical gap in the literature through a comprehensive approach. I have several suggestions that I hope will help further strengthen the manuscript.

Response. We thank you for your positive evaluation. We also appreciate your suggestions for revising the manuscript. In response to your comments, we have made the changes to the manuscript listed below.

1. The authors are encouraged to provide stronger justifications for why it is necessary to single out and study “spiritual boredom” as a distinct type of boredom. The manuscript theorizes that spiritual boredom aligns with the general features of boredom and academic boredom, and the findings support this. However, if spiritual boredom represents the same feeling of boredom manifesting in a different context, further discussion on why studying it as a separate construct is warranted would be helpful. For instance, studying “gaming boredom” or “commuting boredom” would likely yield similar results. While I agree with the authors that it is important to study spiritual boredom, more effort is needed to highlight its unique significance so that readers can fully appreciate its value. For example, academic boredom warrants specific attention because it impacts a distinct population (students), occurs in a more structured and restricted setting (schools), and has unique consequences (e.g., affecting academic performance and learning motivation). Could the authors elaborate on whether spiritual boredom has similarly distinctive characteristics, affects a specific population, and/or carries unique implications?

Response. Following your very helpful suggestion, we outline the characteristics of boredom in terms of population, setting, and consequences in the revised manuscript.

[Spiritual Boredom / Definition]

Spiritual boredom differs from other types of boredom (e.g., academic boredom) in terms of (a) the population experiencing it, which includes people who seek spiritual development and often search for greater meaning in life; (b) the settings, which typically are “silent” environments where spirituality can be experienced and where visits usually are voluntary; and (c) its consequences in terms of a reduction in motivation for spiritual practice and, consequently, spiritual growth. (p. 7)

2. Relatedly, further clarification on how spiritual boredom differs from boredom in other contexts, such as academic boredom or boredom in general, would be valuable.

Response. We hope that the paragraph we have added in response to your comment #1 clarified how spiritual boredom differs from boredom experienced in other contexts.

3. Given the manuscript's focus on the Control-Value Theory, elaborating on why this framework—originally developed to explain emotions in academic and achievement settings—serves as an appropriate lens for understanding boredom in spiritual contexts would strengthen the discussion.

Response. Thank you for bringing this issue to our attention. We agree that the control-value theory was originally developed with a focus on academic emotions. However, from a theoretical perspective and in line with findings in other contexts, the theory has been shown to explain the antecedents and effects of emotions (including boredom) beyond the academic context (Pekrun, 2024). We have made this point more explicit in our revised manuscript.

[Implications for Research and Practice]

Our findings indicate that the CVT, originally developed in the context of achievement emotions, serves as an appropriate theoretical framework for explaining the antecedents and effects of spiritual boredom. Consistent with ongoing theoretical developments suggesting that CVT is applicable beyond academic contexts (Pekrun, 2024), our findings encourage the use of CVT for further research on spiritual boredom (e.g., exploring additional effects of spiritual boredom, such as its impact on the quality of spiritual practice). (p. 34)

4. While acknowledging that research on boredom in spiritual contexts is limited, it would be helpful to include a review of relevant literature on how constructs like meaning, perceived control and motivation the engage relate to spiritual experiences.

Response. Thank you for this suggestion. In the "Previous Research - Literature Search" section of our manuscript, we note: "None of the articles retrieved met any of these criteria. For example, the studies we found looked at whether spiritual or religious people were less bored than others (e.g., Van Tilburg et al., 2019)." We think it would be beyond the scope of our study to outline the relationships of perceived value, perceived control, and motivation with spiritual experiences. Although this is a very interesting and important topic, it is rather different from the focus of our work. For this reason, we have decided not to discuss this line of research in our manuscript.

5. Please provide more methodological details for the studies. For example:

- What exclusion criteria were applied across the studies?
- Were power analyses conducted to ensure adequate sample sizes?
- What information was provided to participants about the studies during recruitment?
- Did the authors collect information about participants' engagement in spiritual practices, such as the frequency or duration of their practice or their years of experience? Providing such details would offer important context for interpreting the findings.
- Please provide more information on the scale development process, as well as the scales' reliability and validity.

Response. Thank you for these helpful suggestions.

5a – exclusion criteria. In the original version of our manuscript, we have noted (p. 15): "For the trait assessments, we aimed to recruit participants with experience in the spiritual practice being studied (i.e., former and current practitioners). For the state assessments, we sought participants

who were currently practicing the respective spiritual practice. [...] In all 10 studies, participants had to be at least 18 years old to take part. Participants had to provide their consent before proceeding with the questionnaire, which began with demographic information, followed by the assessment of all other variables.” There were no further exclusion criteria, which is now highlighted in the manuscript.

[Procedure]

There were no exclusion criteria other than not having experience in spiritual practice, being under 18 years of age, and not giving consent. (p. 16)

5b – power analyses. We did not perform power analyses and point out this as a limitation of our work in the revised manuscript. However, based on your comment, we tried to check the appropriateness of our samples. Given that the correlations of underchallenge, overchallenge and motivation with boredom in the academic context are approximately $|r| = .40$ and that a similar correlation could be hypothesised for the spiritual context (which was indeed the case), the sample size required to demonstrate a hypothesised effect of this magnitude would have been $n = 37$, assuming a statistical power of $1 - \beta = .80$ and an alpha level of $\alpha = .05$. All samples in our 10 studies exceeded this threshold and ranged from $n = 48$ to $n = 414$.

[Limitations]

Fourthly, we recruited participants for our study using snowball sampling and aimed to encourage as many individuals as possible to take part. Future studies on spiritual boredom could perform power analyses to ensure adequate sample sizes. Our initial findings on spiritual boredom may serve as a helpful foundation for such analyses. (p. 31)

5c – recruitment information provided to participants. We have added this information to the “Procedure” section.

[Procedure]

In this recruitment process, studies were labelled as research investigating emotions in the respective context (i.e., yoga, meditation, silence retreat, Catholic sermons, pilgrimage). (p. 15)

5d – information about participants’ engagement in spiritual practices. In each of the 10 studies, we asked participants about their engagement in spiritual practice. We agree that this information may be helpful to readers who are interested in this contextual information. We have included this information in the online supplemental material and refer to it in the revised manuscript.

[Discussion – Occurrence of Spiritual Boredom]

For those who wish to interpret the mean levels of spiritual boredom within specific spiritual contexts in relation to participants’ engagement in spiritual practices, we provide descriptive statistics on participants’ current spiritual practices across all 10 studies in the online supplementary material (SM8). (p. 28)

[SM8: Descriptive Statistics on Participants' Current Spiritual Practices]

Spiritual Context	Study	Trait/State	Variable	M	SD
Yoga	1	Trait	^a Frequency of yoga practice	1.92	0.83
	2	State	^b Frequency of yoga practice	1.72	0.45
Meditation	3	Trait	^a Frequency of meditation	1.53	0.64
	4	State	^b Frequency of meditation	1.48	0.50
Silence retreat	5	Trait	No. of previous silence retreats	6.90	6.40
	6	State	No. of previous silence retreats	14.21	27.25
Sermon	7	Trait	^c Frequency of service attendance	1.32	0.64
	8	State	^c Frequency of service attendance	2.54	0.69
Pilgrimage	9	Trait	No. of previous pilgrimages	5.99	4.03
	10	State	No. of previous pilgrimages	7.70	8.51

Note. a) Participants responded how often they practiced yoga/meditation on a scale ranging from 1 to 3 (1 = *yes, regularly*; 2 = *yes, but not regularly*; 3 = *not currently*). b) Participants responded how often they practiced yoga/meditation on a scale ranging from 1 to 2 (1 = *yes, regularly*; 2 = *yes, but not regularly*). c) Participants responded how often they attended church services on a scale ranging from 1 to 3 (1 = *once every week*; 2 = *multiple times per year, but less than once every week*; 3 = *once per year or less*).

5e - information on the scale development process / scale reliability and validity. Based on your comment, we have now included more information about the development of our spiritual boredom scales, which are at the center of our study. In addition, we now explicitly state that the reliabilities of these scales are shown in Table 2 (Methods section; Measures – Spiritual Boredom). The reliabilities of all other scales are also reported in the Methods section (Measures). Regarding the validity of the spiritual boredom scales, we now refer to this aspect in the Discussion section and highlight the convergent validity of the scales, as indicated by the strong correlations between the MSBS-SF and the Yoga Boredom Scale – State and between the MSBS-SF and the Meditation Boredom Scale - State.

[Measures – Spiritual Boredom]

As the spiritual contexts of yoga, meditation, silence retreat, and pilgrimage consist of different typical facets, we developed scales including items assessing boredom related to these facets based on the approach by Daschmann et al. (2011), which suggests an assessment of situational facets (e.g., for yoga: physical experiences, breathing exercises, relaxation phase,...). However, as sermons do not contain such typical elements, we developed a scale for this context based on the Achievement Emotions Questionnaire (AEQ; Pekrun et al., 2011), which takes into account different components of boredom as outlined earlier (i.e., affective, cognitive, motivational, and physiological/expressive components). (pp. 16, 17)

[Measures – Spiritual Boredom]

We developed a total of 10 scales to assess trait and state boredom in each of the five spiritual contexts that were addressed in our research (see overview in Table 2; reliabilities of all spiritual boredom scales are shown in this table). (p. 16)

[Implications for Research and Practice]

Our initial results indicate high convergent validity of the newly developed scale, as evidenced by the strong correlations between the MSBS-SF (i.e., the short form of the Multidimensional State Boredom Scale; Fahlman et al., 2013) and the YBS-S, as well as the MSBS-SF and the MBS-S. (p. 35)

6. Regarding the state measures, were participants surveyed immediately after engaging in spiritual practices? Also, were all participants currently practicing spiritual practices? If so, this could be a limitation of the research, as those who find spiritual practices particularly boring may have already quit and were not included in the study.

Response. For the state assessments, all participants were asked to complete the questionnaires immediately after engaging in the respective spiritual practice. We describe this more clearly in the revised manuscript. For the trait assessments, we recruited participants with experience of the spiritual practice, that is, both former and current practitioners (see Method section of the manuscript). We consider it a strength of our study that we also included participants who had experience of the spiritual practice but were not currently practicing.

[Procedure]

As for the state assessments, participants were asked to complete the questionnaires immediately after engaging in the spiritual practices. (p. 16)

I hope these suggestions are helpful in further refining the manuscript and strengthening its contribution to the field.

Response. Thank you once again for your very helpful suggestions, which we believe have significantly improved our manuscript.